# A Review of Generators and Power Converters for Multi-MW Wind Energy Conversion Systems

**Saravanakumar Rajendran** [1,*], **Matias Diaz** [1], **Roberto Cárdenas** [2], **Enrique Espina** [1], **Emilio Contreras** [1] **and Jose Rodriguez** [3]

1 Electrical Engineering Department, University of Santiago of Chile, Santiago 9170125, Chile
2 Electrical Engineering Department, University of Chile, Santiago 8370451, Chile
3 Engineering Faculty, Universidad San Sebastian, Santiago 4080871, Chile
* Correspondence: saravanakumar.rajendran@usach.cl

**Abstract:** The rated power of wind turbines has consistently enlarged as large installations can reduce energy production costs. Multi-megawatt wind turbines are frequently used in offshore and onshore facilities, and today is possible to find wind turbines rated over 15 MW. New developments in generators and power converters for multi-MW wind turbines are needed, as the trend toward upscaling the dimensions of wind turbines is expected to continue. Therefore, this paper provides a detailed review of commercially available and recently proposed multi-MW wind turbine generators and power converters. Furthermore, comparative analyses indicate the advantages and disadvantages of commercially available and promising technologies for generators and power converters at the multi-MW target.

**Keywords:** generators; power converters; wind turbine; onshore and offshore wind turbine

## 1. Introduction

Renewable energy sources have become one of the most attractive alternatives to lessening the consequences of global warming. Accordingly, renewable energy sources have been expanding worldwide [1]. As a result, the total global renewable energy capacity has increased from 1331 GW in 2011 to 3068 GW in 2021 [2]. Furthermore, renewable capacity is predicted to maintain its constant growth, accounting for almost 95% of new power installations, averaging about 305 GW per year between 2021 and 2026 [3].

Wind energy has the fastest and most relevant evolution out of all renewable energy sources. At the end of 2020, the world's total installed wind power capacity reached 743 GW, with 93 GW being installed in 2020 [4,5]. By 2021, the installed capacity of global wind energy exceeded 840 GW, driven by an unprecedented expansion in China that exceeded 47.6 GW [6]. According to the International Energy Agency, wind energy will keep expanding as 160 GW of new Wind Turbine (WT) installations are expected by 2025, and 280 GW by 2030 [7].

The power that can be extracted from the wind depends on the size of the turbine, the length of its blades, and it is proportionate to the cube of the wind velocity [8]. Therefore, the development of Wind Energy Conversion Systems (WECSs) has relied on upscaling WT dimensions and looking at installations with higher wind speeds. On the one hand, WTs have reached WT diameters over 170 m [9], and companies such as Siemens, General Electric, Bewind and Mingyang have WTs models for power ratios above 10 MW [5] as shown in Figure 1. Currently, Mingyang MySE 16.0-242 is the world's largest single WT with a rating of 16 MW [10]. On the other hand, offshore technology has rapidly evolved as offshore installations can produce more energy due to the increased availability of wind resources. As a result, the offshore worldwide installed capacity is expected to reach 134 GW in 2026 [3].

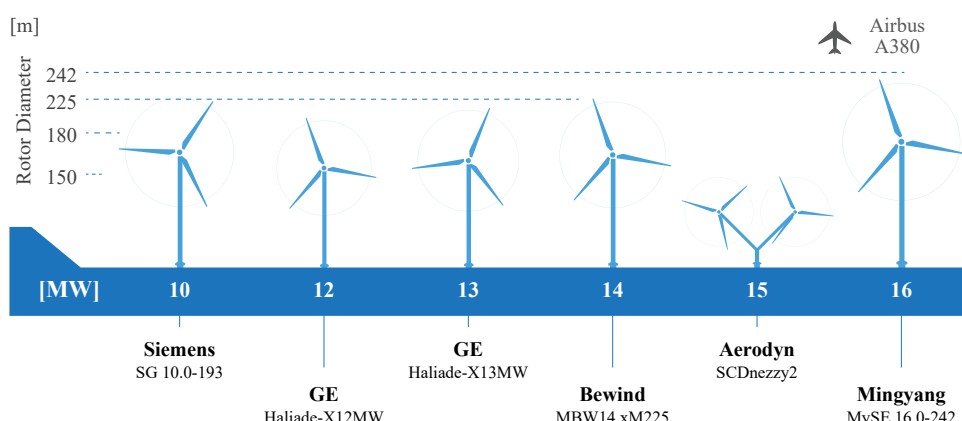

**Figure 1.** Multi-MW WT models.

In addition to the incipient development of multi-MW WECSs at the industrial level, a great deal of academic research is being carried out to enhance the operation of multi-MW WTs, especially for generators and power converters. For example, detailed analyses of electrical generators are presented in [11–14], recommending Permanent Magnet Synchronous Generators (PMSGs) [15], and Doubly-fed Induction Generators (DFIGs) as the leading WT generator technologies. For higher up to 2.5 MW, [16] presents an extensive investigation of WT generators and their market trends, concluding that superconducting generators could replace Permanent Magnet Synchronous Generators (PMSGs). Detailed technical reviews of WECS and driving train topology with Maximum PowerPoint Tracking (MPPT) techniques are presented in [17]. Also, WECSs are compared based on volume, weight, cost, efficiency, system reliability, and fault ride-through capability. Finally, four different MPPT techniques are compared, and modifications for each method are discussed. Recent improvements in WECS, wind farm-related issues, and a review of a critical component in WTs have been discussed and analyzed in [18,19]. The unpredictable nature of the wind causes the following problems in wind power systems: voltage instability, frequency oscillation, and small signal stability issues. Therefore, the transient stability of large-scale wind farms with DFIG WT has been analyzed [20]. In addition, the impact of various faults in power systems has been studied in [21]. The transient response of large-scale offshore WTs has been investigated in the presence of symmetrical and unsymmetrical disturbances [22]. An overview of power converters and the technical requirements for MW WECSs are discussed in [23]. Furthermore, the authors studied the impact of the full-scale converter in PMSG WT [24].

Different power converter topologies applied to permanent magnet generators, Induction Generators (IGs), Synchronous Generators (SGs), and DFIGs with control schemes are discussed in [11,13]. A comprehensive analysis of various power converter topologies, grid-connected wind farms, and fault ride-through methods for PMSGs is discussed in [25], which predicts that PMSG will be dominant in the future wind industry. The design of the multiphase generator and the multiphase converter topology for the WT has been studied in [15]. The overview of power converters and the technical requirements for MW WECSs are discussed in [23].

This study focuses on the current trends in generators and power converters solely used for multi-MW WTs (above 6 MW). In summary, the main contributions of this work are as follows.

- Multi-MW WT generators for onshore/offshore WTs reported in the literature are discussed in the paper with benchmarks based on technological trends and market penetration.

- A detailed comparative study of WT generators is discussed in Section 3.7, and the commercially available generators for different manufacturers are presented and discussed.
- The future trend for WT generators is discussed (ref Section 4), and the high-power generators under the development stage are also presented.
- In addition, a broad range of power converters employed for multi-MW WT generators is presented in this article, with benchmarks focused on technological and market status.
- A detailed comparative study of the different converters and future trends for power converters are also presented.

The paper is organized as follows. First, the overview of WECSs is discussed in Section 2 and then, different types of generators employed in multi-MW WECS are discussed in Section 3. Next, Section 4 presents recent trends in multi-MW generators. Furthermore, Section 5 describes recent power converters used for multi-MW WTs, and finally, in Section 6, conclusions and recommendations are drawn, and future trends are illustrated.

## 2. Wind Energy Conversion Systems

WECS size and functionalities have increased over the past few decades. As presented in Figure 2, a modern WECS comprises a wind turbine, a gearbox, an electric generator and a power converter. The first generation of WT was known as Type I WECS or "Danish" WT for fixed-speed operation. Most of the time, the electric generator was directly connected to the grid. Later, limited variable speed WECSs was developed, known as Type II WECS. Currently, state-of-the-art WECSs can provide full variable speed operation and several control functionalities. Figure 2 shows a Type III WECS when using DFIG or a Type IV WECS when using IG, PMSG, or other generators. The main requirements for a WECS control system are summarized in Figure 2 as follows:

- Basic control functions
- WECS-specific control functions
- Grid services

Basic control functions, such as voltage/current controllers and grid synchronization, guarantee the proper operation of the power converters and maintain the voltage and frequency in the grid, respectively. Furthermore, specific control functions are divided into MPPT/power limitations and fault ride-through. The wind speed is classified according to its regions, such as region 1, region 2 and region 3. Region 1 refers to low wind speed. In Region 2, the torque control extracts the maximum power from the wind at a wind speed below the nominal rate [26,27]. In Region 3, a pitch controller maintains constant power at high wind speed [28,29]. In addition, a Low Voltage Ride-Through (LVRT) has enabled the retention of the WECS in the utility network under low voltage conditions [30,31]. Generally, a grid-forming WECS controller considers the nominal voltage and frequency as a reference signal, and it is also called V-f mode. As mentioned in [32], the V-f controller provides a low output impedance, and for parallel operation with other WECSs, it demands synchronization modules. However, these types of structures involve additional costs to WT. Therefore, to avoid this issue, decentralized droop control is employed to control the parallel converters in an autonomous grid [33]. Finally, other auxiliary services such as droop control [34] and synthetic inertia control [35] are adapted to improve frequency drop and system stability.

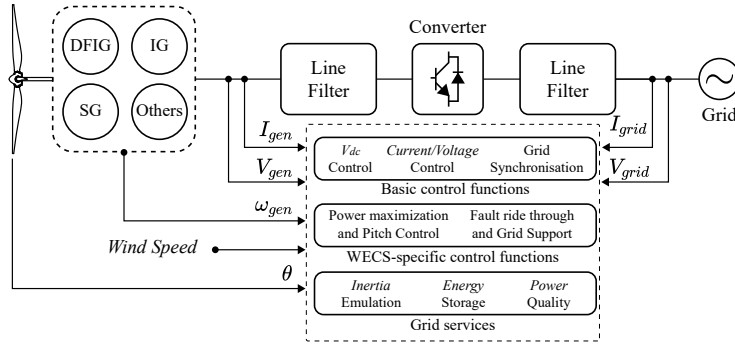

**Figure 2.** Overall control requirements for grid-connected WECS.

## 3. Generators for MW-WECS

Several alternatives used to implement multi-MW variable speed WECS are illustrated in Figure 2. Three typical generators used in this power range are IG, DFIG, and SG [36–39]. The latter could be either the PMSG or the Wound Rotor Synchronous Generator (WRSG). This section illustrates conventional and recent generators utilized in high-power WECS.

### 3.1. Induction Generator

The Squirrel Cage Induction Generator (SQIG) is a simple and robust machine. The cage machine is inherently brushless (unlike the DFIG) and requires reduced maintenance [36]. For variable speed operation, the IG is interfaced to the grid using two full-power converters allowing good fault ride-through capability with this topology. Because the IG is an induction machine, constructing a low-speed multi-pole machine is not technically feasible for direct drive operation [40]. Further, the variable speed operation does not exist in IGs (as in the DFIG).

### 3.2. Doubly-Fed Induction Generator

DFIGs are widely employed for wind energy applications [37]. For a typical DFIG, the rotor is connected to the power converter, and the speed range is restricted, e.g., typically a 30% of this value.

For DFIGs-based WECSs, majorly 3-stage gearboxes are usually required because of design problems associated with the implementation of multi-pole low-speed DFIGs [40]. However, the design of a WECS based on DFIG with a single stage gearbox is presented in [40], and the design of a direct drive multi-MW DFIG is discussed in [41]. Unfortunately, commercial implementations of WECSs based on DFIGs operating at low speed (direct drive) have not yet been reported.

Nowadays, the topology based on DFIGs with partial-scale power converters is still widely used in wind energy applications. However, the difficulties associated with fulfilling the new stringent grid codes of several countries [42]. Therefore, in the future, the preferred multi-MW WECS will be based on synchronous generators (either PMSG or WRSG) interfaced to the grid using full-scale power converters [23,43]. Sinovel (SL6000/128 and SL6000/155), Senvion (6.2M126), United Power (UP6000-136) and Ingeteam are commercial DFIG manufacturers with a power rating of 6 MW and above.

Two of the frequently mentioned disadvantages of the doubly-fed induction generators are

- The conventional DFIG requires slip rings and brushes to connect the rotor to the power converter. This produces well-known issues associated with maintenance and robustness.
- The hardware and control systems required to achieve fault-ride-through capability in DFIG-based WECS are relatively complex.

However, to overcome the above disadvantages, the following doubly-fed generators such as Brushless Doubly-Fed Induction Generator (BDFIG), Brushless Doubly-Fed

Reluctance Generator (BDFRG), Brushless Cascade Doubly Fed Induction Generator (BCD-FIG) and Dual-Stator Brushless Doubly-Fed Induction Generator (DSBDFIG) have been proposed in the literature.

### 3.2.1. Brushless Doubly-Fed Induction Generator

A WECS based on BDFIG is shown in Figure 3a, and it is made up of two stators wounded to be magnetically independent between them. The two independent stators are called "Power winding" and "Control winding". In addition, the control winding is designed to supply a fraction of the nominal power. Usually, a back-to-back (BTB) power converter connects the control winding and the grid. In addition to brushless operation capability and robust rotor construction, one of the advantages of this configuration is the ability to ride through faults without crowbars [44]. However, the main drawback of the BDFIG is the torque density; that is, this machine produces less torque per volume than the DFIG.

### 3.2.2. Brushless Doubly-Fed Reluctance Generator

The BDFRG consists of two sets of three-phase winding, such as primary and secondary winding [45,46]. In WECS, the primary winding is directly connected to the grid, whereas the secondary winding is connected through the grid via a partial scale converter. Figure 3b shows a WECS based on BDFRG. The reliability of the BDFRG has been increased due to the brushless construction. In addition, the fault ride-through capability has been improved because of the high leakage inductance presented in the stator winding of BDFRG [47].

### 3.2.3. Brushless Cascade Doubly Fed Induction Generator

Conventional DFIG requires brush wear and maintenance of carbon accumulation, leading to additional maintenance costs and less reliability. BCDFIG can overcome the above demerits [48]. A cascade induction machine combines two wound rotors, i.e., a Permanent Machine (PM) and a Control Machine (CM). The pole pairs for PM and CM are $p_1$ and $p_2$ respectively. Figure 3c shows the schematic of BCDFIG. The brushes are eliminated by coupling both machines mechanically and electrically via rotors. A BCDFIG wind turbine is connected to a gearbox, and the generator variable speed range determines the gear ratio, i.e., $\pm 30\%$. The number of poles is increased in BCDFIG, which reduces the gear ratio; subsequently, the size and cost of a gearbox are reduced. In BCDFIG, the converter directly connects to the control machine and improves the transient behaviour.

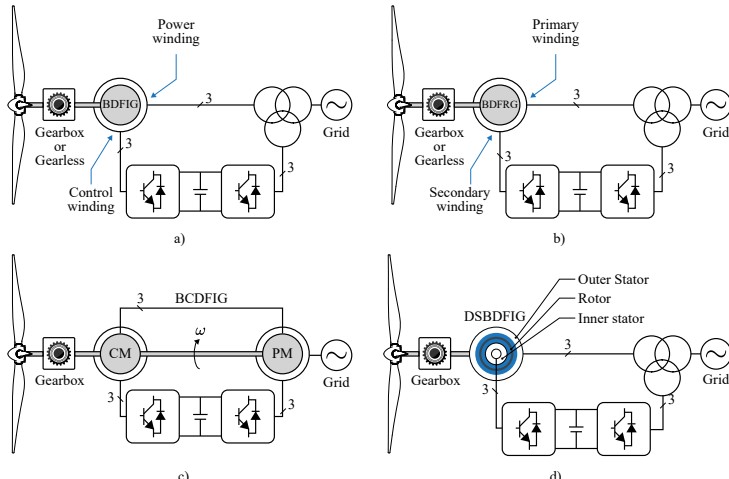

**Figure 3.** Different types of DFIG WECS configurations. (**a**) BDFIG. (**b**) BDFRG. (**c**) BCDFIG. (**d**) DSBDFIG.

### 3.2.4. Dual-Stator Brushless Doubly-Fed Induction Generator

Generally, brushless DFIG has direct coupling between stator fields, and it introduces inevitable harmonics [49]. To overcome these effects, a novel DSBDFIG is proposed [49,50]. Figure 3d shows the schematic of DSBDFIG. It consists of three parts, i.e., the outer and inner stator with three-phase balanced winding and nonmagnetic support with a dual-layer reversely connected to balanced three-phase winding. It has a compact structure with a lower gear ratio which helps to reduce the size of the gearbox. Wind generation based on DSBDFIG requires a partially rated converter, which increases system reliability and efficiency.

The above-discussed generators, such as BDFIG, BDFRG, BCDFIG and DSBDFIG, are conceptually well-established techniques but not commercially available.

### 3.3. Synchronous Generators

The SGs, particularly the PMSGs, are considered essential technologies for implementing WECSs. There are several commercial solutions based on this generator for low-speed operation (direct-drive) or medium-speed operation ("Multibrid" concept with reduced-size gearbox [51]). Some examples are the Enercon E-126 of 7.58 MW equipped with a synchronous annular generator; and the Siemens SWT-8.0-154 (PMSG) of 8 MW. There are several advantages to using SG-based WECSs. For instance, good fault ride-through capability is provided by the full-scale power converters [43]. Moreover, neither slip-ring nor brushes are required in a typical PMSG, and direct drive operation reduces the audible noise because the gearbox is eliminated from WECSs. The weight, cost, and efficiency of direct-drive WRSGs, direct-drive PMSGs, and three-stage geared DFIGs are compared in [37,40]. Generally, the PMSG has divided into two categories based on the direction of the magnetic flux crossing in the airgap: Radial Flux (RF) and Axial Flux (AF) machines [52]. However, due to economic constraints, most commercially available PMSGs are RF machines [53]. In addition, this topology provides robust design and high structural stability. Siemens already has commercially existing RF-PMSG, such as SG 11.0-200 DD (11 MW), and a 14 MW DD-RF-PMSG is under development. In addition, the Haliade-X 12 MW from GE developed an offshore RF-PMSG with DNV-GL certification. The AF-PMSGs have been investigated for small and medium WTs. Detailed examination of AF-PMSG-based wind generators for offshore applications has been studied in [54] for the range of 3 to 12 MW. Furthermore, a 10 MW iron-less AF-PMSG has been discussed in [53] for offshore application. This analysis concluded that the overall weight of the machine is reduced because of the ironless generator. Finally, the commercial availability of the AF-PMSGs is in the development stage. Overall, a PMSG-WECS heavily depends on the price of the rare earth elements required to fabricate the permanent magnets. This commodity has suffered large price fluctuations in the past years; for instance, in 2021, the cost of neodymium increased about 78 times compared to the cost in January 2015 [55].

Aerodyn, Goldwind, and MingYang have manufactured synchronous generators with more than 7 MW power ratings. Nevertheless, Enercon (the 7.5 MW E-126) and Aerodyn (SCD nezzy2 twin rotor) have successfully used this technology (electrical excitation) to produce one of the largest and most reliable WECS solutions available in the WT market.

### 3.4. xDFM

The xDFM is a new WECS topology proposed in [56] and is marketed by the Spanish company Ingeteam for onshore and offshore applications with power ranges up to 6 MW [56].

The proposed "xDFM" topology is shown in Figure 4a. It is based on a permanent magnet machine, and a DFIG affixed to the same mechanical shaft. The BTB converters are connected to a PM machine typically operated as a generator below synchronous speed and as a motor above the synchronous velocity, feeding a fraction of the power back to the shaft. The rating of the PM machine is reduced, but, on the other hand, the DFIG-stator must be designed to deliver nominal power. According to [56], the main advantage of

the xDFM, compared to the conventional DFIG, is the much enhanced LVRT capability. Another advantage of xDFIM is that DFIG-stator windings could be designed for medium voltage operation, i.e., reducing the size of the transformer required for grid connection. Moreover, the PM machine can be controlled using standard field-oriented techniques to reduce the torque peaks and oscillations produced in the mechanical shaft during faults. Furthermore, it is claimed that power smoothing of the generated power could be achieved using the proposed topology.

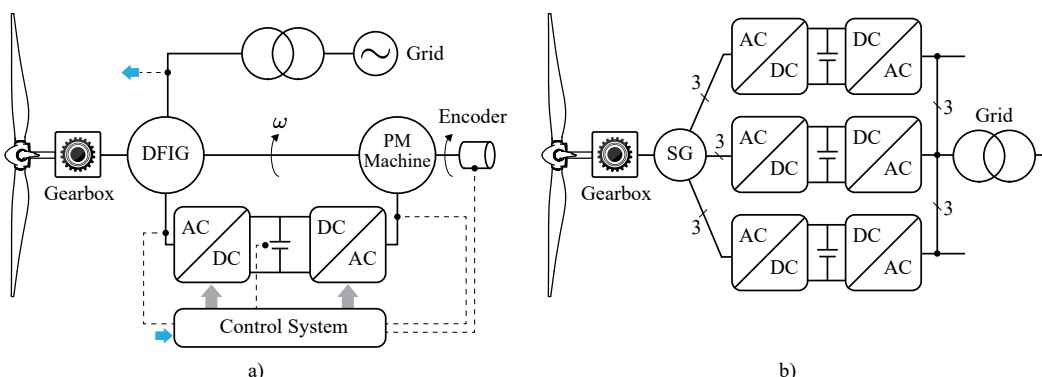

**Figure 4.** WECS configurations. (**a**) xDFM. (**b**) Multi-channel synchronous generator.

### 3.5. Superconducting Generators

Superconducting Generators (SCG) offer advantages such as high efficiency, low maintenance, and high-power density. Due to the high current density, it is feasible to reduce the weight and volume of the superconductor generators by about 40% [57]. Therefore, superconducting generators offer an up-and-coming solution for high-power WECSs rated at 10 MW or above.

Nowadays, three main superconducting wires are available in the market: High-Temperature superconductors (HTS), Low-Temperature superconductors (LTS), and Magnesium diboride ($MgB_2$). A comparative study of HTS and LTS generators for WECS applications has been carried out in [57]. This study demonstrates that development in HTS generators is more significant than LTS generators. However, the high cost of super-conductive wires, along with cryogenic and refrigeration systems, are the main drawbacks of the commercialization of SCGs in the wind power sector [58].

#### 3.5.1. High-Temperature Superconducting Generator

An HTS generator achieves a high power density for multi-MW applications [59,60]. The high current densities reduce the mass and volume of HTS generators by about 40–50%. Moreover, the generator losses could be halved. A cooling system is required to maintain the temperature in the superconducting winding. Therefore, it reduces efficiency and increases complexity. For example, a typical 4.5 MW generator operates at 30° K, requiring 0.16% of total power to cool [60]. Detailed information on the mass and volume reduction attainable in the generator, nacelle, tower, etc. of WECSs based on the HTS generator is available in the report published by the National Renewable Energy Laboratory (NREL) in 2010 [61]. In addition, WindTec has a 10 MW direct drive WECS (the SeaTitan) made up of an HTS generator [62]. The EU-funded project Suprapower (see [62] ) also seeks to develop WECS of 10 MW or more using HTS technology. Converteam [63], and Changwon National University [64] are developing a second generation of HTS-based wind turbine generators.

#### 3.5.2. Low-Temperature Superconducting Generator

A detailed study of 12 MW, SuperConducting wind generator (SCWG) with LTS field winding for offshore WT has been discussed in [65]. A comparison has been made between SCWG and PMSG on the basics of cost and weight. This analysis concludes that SCWG is

46% lighter than PMSG. In addition, a weight-to-power ratio is compared with different SCWGs like AMSC, GE, and TECNALIA.

General Electric Research of Niskayuna, Newyork, developed an LTS generator with high-efficiency ultra-lightweight [66,67].

### 3.5.3. Magnesium Diboride

A 10 MW, 8.1 rpm direct drive partially SCG has been developed using $MgB_2$ as a field coil [68]. It consists of a warm rotor with a superconducting coil, which works at 20 K. A cryocooler is installed on the rotor, which extracts the heat from the superconducting coil. As a result, the SCG has a 26% and 11% reduction in weight and a lighter tower than a permanent magnet generator, respectively. A 20 MW superconducting synchronous generator is designed using $MgB_2$ superconductor on both stator and rotor windings [69]. The stator and rotor windings operate at 10 K and 20 K, respectively. Two separate cryogenic systems are considered to improve system reliability. The proposed superconducting generator is 2.5 times lighter than PMSG, reducing the cost of the tower and foundation; also, the levelized cost of energy (LCOE) is reduced by 8.5%. In addition, the $MgB_2$ is an alternative for HTS and LTS because the cost is lower than HTS and the cooling method is more superficial than LTS [70].

Furthermore, HTS-based SCWGs are commercially available. Moreover, the remaining two, that is, LTS and $MgB_2$, are under the conceptual level.

### 3.6. Multi Channel Generator

A fault-tolerant WECS could be implemented using a generator with several separate and isolated stator windings, each feeding a BTB converter. This solution has shown in Figure 4b. The main advantage of the multichannel topology is fault redundancy [71,72]. Suppose a fault is created in one of the phases; the generator could be designed to maintain nominal power operation even in one phase failure. It is recommended to design the machine with a phase inductance of 1.0 p.u to limit the internal failure [72]. Moreover, to increase the fault tolerance capacity of the system, a multiple-window transformer could be used [23]. The multi-channel generator is used in commercial solutions, for instance, the Gamesa 10x WECSs of 4.5 MW and 5 MW.

### 3.7. Comparison of Multi-MW WECS

Table 1 shows the pros and cons of different wind turbine generators. Several researchers have compared various types of wind turbine generators [40,73–77]. Five different multi-MW generators, such as Doubly-Fed Induction Generators with a three-stage Gearbox (DFIG3G), Direct-Drive Synchronous Generator (DDSG) with electrical excitation, Direct-Drive Permanent-Magnet Generator (DDPMG), Permanent-Magnet generator with single-stage gearbox (PMG1G), and Doubly-Fed Induction Generator with a single-stage Gearbox (DFIG1G) are compared based on the cost and annual energy yield without considering the integral parts of the installation of turbines [40]. DFIG3G is lighter and widely used; at the same time, low energy yield due to the gearbox. DDSG and DDPMG are expensive alternative solutions for high-power WECS. DFIG1G offers a higher energy yield per cost, but it is too special for manufacturers. On the other hand, a DDPMG is considered an optimal generator for the fully rated converter due to the absence of a gearbox. Four different generators are compared based on cost, efficiency, power consumption, topology, and control complexity [73,74]. This study demonstrates that a permanent magnet generator with a fully rated converter is more appropriate for offshore wind locations. The replacement of an asynchronous generator with a synchronous generator is discussed in [75]. This study deduces that replacing induction with synchronous is appropriate for a power rating of less than 750 kW machines. Various ratings of large multi-MW DDPMG are compared based on mass, mass per torque, and active & inactive materials [76]. This study discloses that a shorter flux path is required to minimize the active material of DDPMG.

Commercially available multi-MW generators are compared based on generator topology with their connections to the turbine [77]. This study infers that the reliability of the turbine can be improved by eliminating the gearbox and large-scale converters. Three different generators have been compared based on weight, stator radius, power loss, cost estimation, fault recovery, and noise reduction [36]. It shows that Direct Drive (DD) synchronous and permanent magnet generators are suitable for offshore applications because of their robustness and reliability. The geared and direct drive generators are compared on the basis of energy yield, cost, and weight [53]. This study concludes that DD PMSG feature the highest energy density among available technologies.

Economic, technical benefits and limitations for DD PMSG are studied in [78]. The main drawback of this topology is considerable in size and weight. Therefore, logistics and construction are limited in the case of offshore applications. Similarly, high-rated power generators introduce high current density in stator coils and produce more heat. So, proper cooling is necessary to overcome this heat and executed by the liquid cooling method. Different types of wind turbine generators and a comparison of generators based on technical data and commercial availability are presented in [16]. This analysis infers that more than 80% of the generators are DFIG on the market. However, for large-rating turbines, PMSG is a better option. Three different generators have analysed and concluded that geared SCIG is most appropriate for small-scale standalone wind energy systems [79]. SC generators and permanent magnet generators are compared on the basis of weight, volume, and cost in [57]. This study discloses that SC generators are more appropriate in weight, volume, and cost for more than 8 MW. The mass of HTS generators is roughly 50% less than Permanent Magnet Direct Drive (PMDD). In addition, the cost of materials is the main drawback of HTS generators. A detailed study on a constant speed with a squirrel cage induction generator and three variable speed systems with DFIG and DD are discussed in [39]. The BDFIG with enhanced LVRT is a better option for low-cost, high-reliability analogized with DFIG. Also, BDFIG is more suitable for medium-speed generators, but it is slightly oversized due to additional windings.

**Table 1.** Comparison of different generator topologies.

| Generator | Gear Box Type | Advantages | Disadvantage | Comments |
|---|---|---|---|---|
| DFIG | 1G and 3G | • DFIG3Gs are commercially used.<br>• The power rating of the converter is about of rated power and the range of the speed roughly varies from 60% to 110% of rated speed.<br>• Only 30% of the generated power is used for the converter. | • Low energy yield due to high losses in the gearbox.<br>• Grid fault ride through requirements are essential for DFIG. | • Losses are higher due to gearbox.<br>• Addition protection is required. |
| Brushless DFIG | Medium speed | • Low cost constructions and permanent magnets are not required.<br>• Fractional rated converters are used. Since brushes are absent so failure due to brushes are completely removed.<br>• Significantly improved LVRT performance compared with DFIG. | • Its slightly larger due to additional windings | • Commercially not available for multi-MW WT. |
| Synchronous Generator | Direct Drive | • Electrical excitation should be provided either rotor side or permanent magnet. | • Full scale and reduced scale converters are required. | – |
| PM generator | Direct Drive | • Higher efficiency and reliability.<br>• According to generator structure RFPM is quite simple compared to AFPM and TFPM.<br>• Complicated in construction structure<br>• TFPM is more flexible for new technology.<br>• Overall PM are more suitable offshore wind turbine. | • Permanent magnet cost is fluctuating in market.<br>• Rare earth materials are required. | • Most of the offshore wind turbines are DDPMSG.<br>• The rare-earth PM materials are eliminated by electrical and hybrid excitation. |
| High temperature superconducting | Direct Drive | • Low weight, small size and higher efficiency.<br>• SC generators are significantly advanced over DD PMG in terms of shear stress of 53 kPa and efficiency of 96%.<br>• HTS concepts are high superior than<br><br>DDPM for more than 8 MW. Rotating field and radial topology concepts are most common in SC. | • Cryogenic cooling system is most essential for SC generator.<br>• Cost of HTS material is about 90% total active material, due to this reason the cost of the generator is very high. | • Technically under development stage.<br>• $MgB_2$ is an alternative for HTS and LTS. |

Various design techniques have been investigated for the weight and cost of HTS generators [58]. Also, this study deals with minimizing the use of the HTS field coil, which reduces the cost of generators. A qualitative comparison for brushless DFIM, BDFRM, and others is presented in [80]. Three types of wind generators are compared, namely DFIG1G, DFIG3G, and DDPMG [81]. This study demonstrates that DDPM generator efficiency is about 96%, and the overall efficiency of the PMSG improvers by optimising its parameters.

SC generators are significantly advanced over PMDD generators in terms of shear stress of 53 kPa and efficiency of 96% [82]. Different types of large wind generators are compared based on direct drive, semi-direct drive, and indirect drive [83]. This analysis found that DD generators are more capable of high-power WECSs. In addition, electrical excitation and hybrid excitation are proposed to eliminate the rare-earth PM material in the case of PMSG. Different drive train topologies of a 10 MW PMSG, namely DDPMSG, medium-speed PMSG, and high-speed PMSG, are extensively compared for offshore WTs [84]. This study claims that gearbox usage reduces the size of WT and raw materials. A comprehensive analysis of the commercial design of electrical generators utilised in the high-power wind industry is presented in [85]. In addition, the performance of the generators is assessed by their mass, cost and mass-to-torque ratio. Finally, this analysis concluded that radial flux machines are appropriate for DD WTs.

## 4. Recent Trends in Generators

This section presents the recent advances in wind turbine generators. Table 2 shows the different topologies of wind generators with a power rating of 6 to 15 MW. High-power WTs are classified mainly into two categories, such as direct and indirect drive trains. Except for HTS and EESG, other topologies prefer the indirect drive train WT (geared) as presented in Table 2. In an indirect drive train, gearbox failure is a crucial parameter for turbine downtime [86]. Specifically, in the case of offshore, this technology is complex and highly expensive. Although DDs have many advantages, certain drawbacks need to be addressed for further development in DD WTs. Rotating velocity decreases at high power levels, introducing a torque increase in DD generators [87]. Furthermore, this increase in torque is directly proportional to the tangential force density and the diameter of the air gap [76]. Thus, a larger diameter with a higher force density is essential to accommodate these changes. However, the above modifications increase the weight and cost of the generator.

Presently, most of the WT generators are DD EESG and DD PMSG. Enercon manufactures the EESG rated at 7.5 MW, which is the updated version of E-112 4.5 MW [88]. In addition, some manufacturers like Lagerwey and Torres adapted EESG, and the rating of the machines varies from fractional kW to 3 MW. However, upscaling these machines is a challenging task and expensive. Due to the above reasons, most manufacturers do not prefer the EESG. Furthermore, the direct drive wind turbine (DDWT) requires a large number of poles, whereas pitched poles are limited EESGs [85]. Hence, high-power, high-torque generators require higher volume and weight, and a complex cooling system is essential to minimize thermal losses. Overall, EESGs are robust and easy to construct, but are the weightiest and most expensive generators compared to other topologies such as DFIG, DDPMSG, and single-stage gear PMSG [40].

DD PMSGs are highly suitable for high-power wind energy applications. Generator reliability has increased due to the elimination of the slip rings, which reduces maintenance costs [40]. In addition, the absence of an external energy source reduces the copper loss. Table 3 shows the list of manufacturers that prefer DD PMSG over other configurations for the above reasons. The following manufacturers, such as Siemens, Vestas, General Electric, MingYang, Goldwind, and Samsung, have DD and geared PMSGs. For example, Siemens has a 10 MW DD PMSG with a rotor diameter of 193 m. Also, Vestas and MingYang have geared and medium-speed PMSGs with a rating of 8 to 10 MW. Table 4 shows the wind turbine generators in the development stage. The details mentioned in Table 2, Table 3 and Table 4 are gathered from the online portals [89,90]. The highest power rating of the WT generators has been manufactured by MingYang and Vestas, with a power rating of 16 MW

and 15 MW, respectively. In addition, General Electric and Siemens develop a 14 MW generator with an annual energy production of approximately 74 GWh and 80 GWh. This table shows that the current trend of the generators is moving towards permanent magnet generators. However, manufacturing these generators demands Rare Earth Materials (REM) like neodymium iron boron or samarium cobalt [85]. Most PMSGs utilize neodymium iron boron as a result of its magnetic property. Therefore, the main drawback of these generators is the REM, even though the technical merits of these generators are well enough. The following alternative solutions may be considered for replacing REM in the wind industry.

- The HTS generator is one of the promising technologies without using REM. In addition, this technology offers lightweight generators with higher efficiency. At present, AMSC has this HTS generator with a power rating of 10 MW.
- The theoretical analysis has been conducted between ferrite magnet-based synchronous generators with conventional PMSG for 6 MW WT with the same stator design [91]. This study concludes that both generators are almost similar in terms of energy cost. However, optimizing the ferrite PMSG would be the alternative for neodymium iron boron-based PMSG (this solution is appropriate when the price of neodymium iron boron is increased continuously).
- REM can be replaced by double excitation [85]. In addition, the radial flux machines have the better option for DD WTs.

**Table 2.** Different topologies of High power WT generators.

| Manufacture | Model | Generator Type | Gear Box | Power (MW)/Rotor Diameter (m)/Speed (rpm)/Voltage (kV) | Onshore or Offshore | Commercial Status |
|---|---|---|---|---|---|---|
| **DFIG Manufacture** | | | | | | |
| Sinovel, China | SL6000/128 SL6000/155 | DFIG | 1-Stage and 2-Stage Planetary | 6MW/128/1200/6.3 | Onshore | Available |
| United Power | UP6000-136 | DFIG | – | 6 MW/136/–/6.6 | Onshore | Available |
| Senvion | 6.2M126 | DFIG | – | 6.15 MW/126/1170/33 | Onshore | Available |
| REpower | 6.2M152 | DFIG | planetary | 6.2 MW/152/–/– | – | Available |
| Ingeteam | – | DFIG | 3-Stage | 9 MW/ | Both | Available |
| **HTS Manufacture** | | | | | | |
| AMSC, USA | wt10000dd | HTS (cryogenic and water cooling) | DD | 10 MW/190/10/12 | Offshore | Available |
| **EESG Manufacture** | | | | | | |
| Enercon | E-126 7.580 | EESG | DD | 7.5 MW/127/12/0.69 | Onshore | Available |
| Aerodyn | SCD 8.0 MW | EESG | DD | 8 MW/168/–/– | – | – |
| **SG Manufacture** | | | | | | |
| Aerodyn, Germany | aerodyn aM 6.0/139 | ASG / PMSG | – | 6MW/139/–/3.3 | – | – |
| | aerodyn SCD 8.0/168 | Synchronous | Planetary | 8 MW/168/308/– | Both | – |
| | aerodyn SCD nezzy2 twin-rotor | synchronous with brushless electrical field excitation | two-stage planetary gearbox with flex pins | 15 MW/150/– | Both | 2022 |
| | aerodyn SCD 8.0/168 | synchronous (electrically excited) | Planetary | 8 MW/168/– | – | – |

**Table 3.** PMSG based commercially available high power WT generators.

| Manufacture | Model | Generator Type | Gear Box | Power (MW)/Rotor Diameter (m)/Speed (rpm)/Voltage (kV) | Onshore or Offshore | Commercial Status |
|---|---|---|---|---|---|---|
| | | **PMSG Manufacture** | | | | |
| Ingeteam | – | PMSG | DD/1G/3G | 9 MW/– | Both | Available |
| Siemens Gamesa | SWT-7.0-154 | PMSG | DD | 7 MW/154/– | Offshore | Available |
| | SWT-7.0-154 | PMSG | DD | 7 MW/154/– | Offshore | Available |
| | SWT-6.0-154 | PMSG | DD | 6 MW/154/– | Offshore | Available |
| | SG 10.0-193 | PMSG | DD | 10 MW/193/– | Offshore | Available |
| | SG14-222 | PMSG | DD | 14 MW/222/– | Offshore | Under development (2024) |
| | SG 11.0-200 | PMSG | DD | 11 MW/200/– | Offshore | Under development (2022) |
| MHI Vestas Offshore, Denmark | V174-9.5 MW | PMSG | Geared (1:41) | 9.5 MW/174/400/– | Both | Available |
| | V164-8.0 MW | PMSG | planetary | 8 MW/164/500/30 | Both | Available |
| | V164-8.3 MW | PMSG | planetary | 8 MW/164/500/66 | Both | – |
| | V164-8.8 MW | PMSG | planetary | 8.8 MW/164/500/30 | Both | Available |
| | V164-10 MW | PMSG | Geared (1:41) | 10 MW/167 | Both | – |
| Swiss Electric, China | YZ127/6.0 YZ140/6.0 YZ160/6.0 | PMSG | DD | 6MW/150/12/3 6MW/170/12/3 6MW/190/12/3 | Both | – |
| | YZ150/10.0 YZ170/10.0 YZ190/10.0 | PMSG | DD | 10MW/127/12/3 10MW/140/12/3 10MW/160/12/3 | Both | – |
| General Electric | Haliade-X 12 MW | PMSG | DD | 12 MW/220/–/6.6 | – | Available |
| | Haliade-X 13 MW | PMSG | DD | 13 MW/220/–/6.6 | – | 2023 |
| | Haliade150-6 MW | PMSG | DD | 6 MW/151/11.5/0.9 | – | Available |
| Goldwind | GW184-6.45 MW | PMSG | DD | 6.5 MW/184/–/ | – | Available |
| | GW175-8 MW | PMSG | DD | 8 MW/175/–/ | – | Available |
| MingYang, China | MySE6.45-180 | PMSG | medium-speed gearbox | 6.45 MW/178/ | Offshore | Available |
| | MySE7.25-158 | PMSG | medium-speed gearbox | 7 MW/158/ | Offshore | Available |
| | MySE8.3-180 | PMSG | medium-speed gearbox | 8.3 MW/178/ | Offshore | Available |
| CSIC, China | MH152-6.2 | PMSG | – | 6.2 MW/152/–/– | Both | – |
| Bewind | BW 6.xM172 | PMSG | 2-stage gearbox | 6 MW/172/–/– | Offshore | Available |
| Dongfang, China | D10000-185 | PMSG | DD | 10 MW/185/10/12 | Both | – |
| | D10000-185 | PMSG | DD | 11 MW/185/10/12 | Both | – |
| | D8000-185 | PMSG | DD | 8 MW/185/–/– | Both | – |
| | D7000-186 | PMSG | DD | 7 MW/186/–/– | Both | – |
| Samsung | S7.0-171 | PMSG | planet flexpin | 7 MW/171.2/400/3.3 | Both | – |
| Sewind, Shanghai | El.W8000-167 | PMSG | DD | 8 MW/167/12/069 | Both | – |

**Table 4.** High power WT generators under development stage

| Manufacture | Model | Generator Type | Gear Box | Power (MW)/Rotor Diameter (m)/Speed (rpm)/Voltage (kV) | Onshore or Offshore | Commercial Status |
|---|---|---|---|---|---|---|
| Siemens Gamesa | SG14-222 | PMSG | DD | 14 MW/222/– | Offshore | 2024 |
| General Electric | Haliade-X 14 MW | PMSG | DD | 14 MW/220/11.5/6.6 | – | 2023 |
| MingYang, China | MySE16.0-242 | PMSG | medium-speed gearbox | 16 MW/242/ | Offshore | 2024 |
| Bewind | BW 14.xM225 | PMSG | 2-stage gearbox | 14 MW/225/–/– | Offshore | – |
| Vestas | V236-15.0 MW | PMSG | medium speed gearbox | 15 MW/236/–/– | Offshore | second half of 2022 |

## 5. Power Converter Topologies for Multi-MW WECS

The WECS power converter has been classified into two categories, i.e., Low voltage (LV) (<1 kV) and Medium voltage (MV) converter (1–35 kV). The LV converters are used up to 3 MW WTs. Different power converter configurations have been developed for a multi-MW WECS. Nowadays, the nominal power of single WT has increased to 10 MW. Due to this, the number of converter modules also increases. A detailed study of LV and MV converters for 6 MW has been studied in [13,92] and the authors concluded that MV converters are more suitable for high power ratings. Figure 5 shows the classification of power converters for multi-MW WECS (DFIG and PMSG).

### 5.1. Parallel Two Level Back-to-Back Converter with Common and Individual dc-Link

Type 3 and Type 4 turbines are configured through BTB common or individual dc-link for DFIG and PMSG generators [93–95]. Two BTB Voltage Source Converter (VSC) configurations are connected in parallel with a common dc-link for a 1.5 to 5 MW power rating. A common dc-link is shared with two BTB converters, which reduces cost and space. The main drawback of this type of configuration is the circulating current that exists both on the generator side and on the grid [93]. The circulating current can be minimized by connecting inductive filters between each converter on the generator side. In addition, Total Harmonic Distortion (THD) at the grid side is eliminated by connecting inductive filter [94]. An individual dc-link resolves the reliability issues. It may lead to higher costs and increase the failure of dc-link capacitors. Figure 6a,b depicts BTB converter for individual and common dc-link, respectively.

### 5.2. Current Source Back-to-Back Converter

Several studies have reported that BTB converters have a significant failure rate in DDWT; also, dc bus electrolytic capacitors are required for special attention among other components [96]. Two Current Source Converters (CSCs) are connected BTB through an inductor [97,98]. It is more suitable for a 5 MW power rating, and the dc-link inductor increases total loss and weight. On the other hand, two VSC are connected BTB through a dc-link which results in the highest maintenance cost [99]. Another disadvantage of CSC is the bulky inductor, which has a slower dynamic response than VSC. Figure 6c shows the BTB CSC.

### 5.3. Neutral Point Clamped Back-to-Back Converter

Two Level (2L) VSCs are placed in this topology by a split dc link capacitor and clamping diode. Figure 6d shows Neutral Point Converter (NPC) for PMSG and DFIG machines. Compared to 2L VSC, it has reduced $\frac{dv}{dt}$ electromagnetic inference. This topology is widely used in MV WTs such as Shandong, XEMC-Darwind, and Zephyros for PMSG. Moreover, the MV stator voltage removes the WT step-up transformer, which is an added benefit of this converter, as well as a considerable reduction in overall cost [100–102].

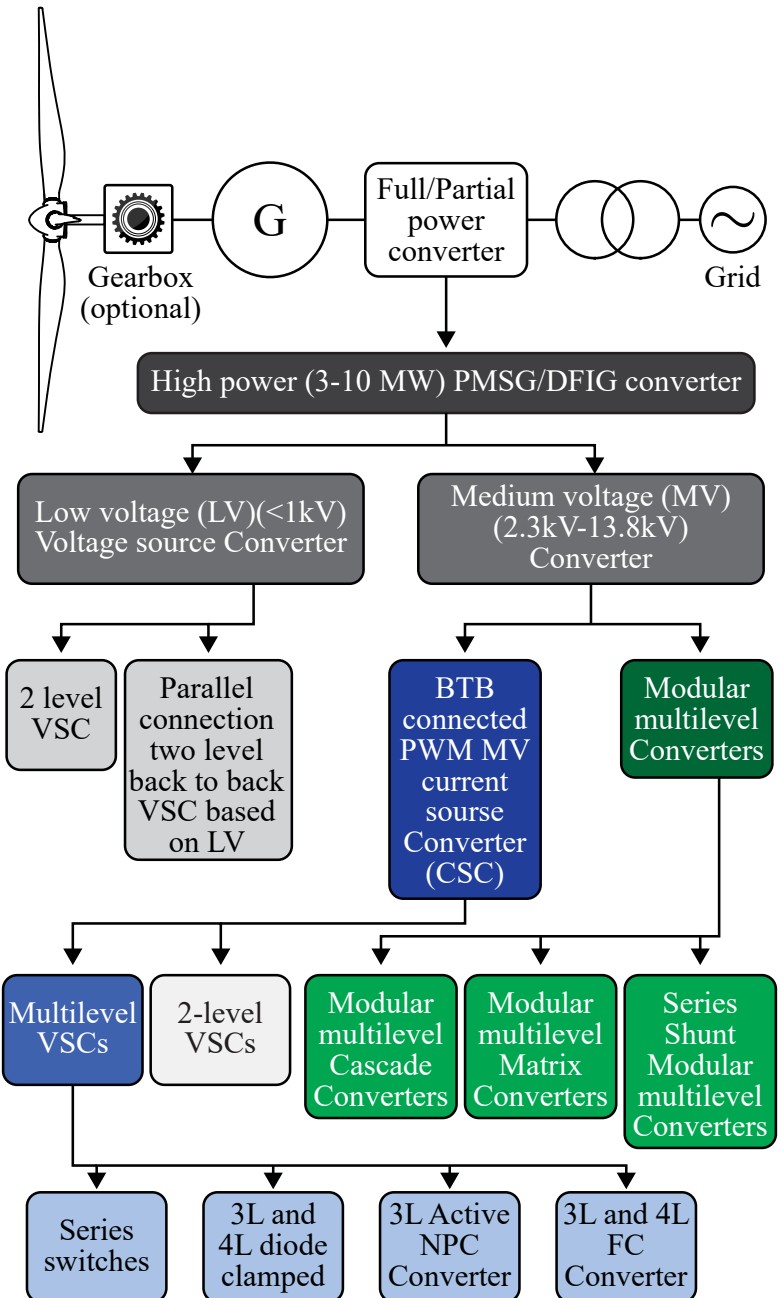

**Figure 5.** Power converters for PMSG and DFIG.

Table 5 shows the commercial availability of power converters for multi-MW WECSs. The parallel 2L BTB and NPC converters are widely used for multi-MW WECSs [92,103]. Most power converters operate in the LV range using semiconductor switches based on LV-IGBT, such as the SINAMICS W180, DFIG 500-5000 and FC LV 100-10000 models. Recently, MV power converters have been widely adopted in the wind turbine industry [104]. For instance, the 15 MW Ingeteam WT (model FC MV 3000-15000) is equipped with a BTB NPC converter. In this case, MV-IGBTs are used to connect the WT to a 3 kV grid. The details mentioned in Table 5 have been gathered from the manufacturer's online portals [105–108].

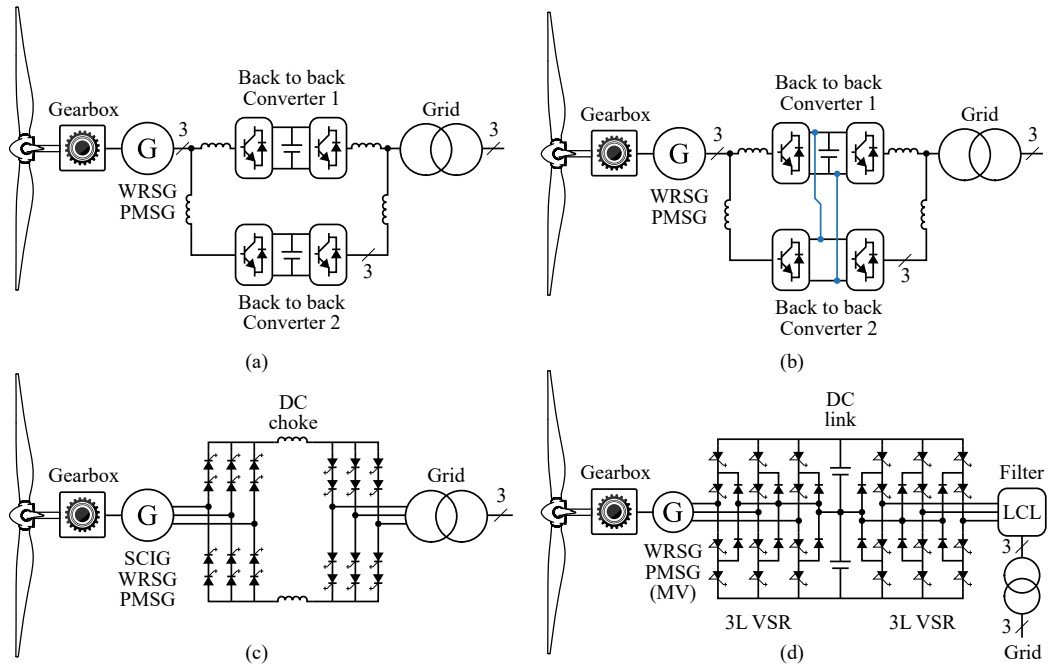

**Figure 6.** (**a**) Parallel BTB converter with individual dc-link. (**b**) Parallel BTB converter with common dc-link. (**c**) Current source BTB converter. (**d**) Neutral point clamped BTB converter.

**Table 5.** List of commercially available power converters.

| Commercial Model | Converter Type | Power Rating | Nominal Voltage | Semiconductor Type |
|---|---|---|---|---|
| PCS6000 | NPC BTB | 4–12 MW | 3.3 kV | IGCTs |
| ACS800-87LC | 2L BTB | 1.5–6 MW | 525–690 V | LV-IGBTs |
| ACS880-87LC | 2L BTB | 1.5–8 MW | – | LV-IGBTs |
| SINAMICS W180 | Parallel 2L BTB | 2 to 10 MW | 690 V | LV-IGBTs |
| DFIG 500-5000 | 2L BTB | 2 MW–5 MW | 690 V | LV-IGBTs |
| FC LV 100-10000 | 2L BTB | 2 MW–5 MW | 690 V | LV-IGBTs |
| FC MV 3000-15000 | NPC | 7.5–15 MW | 3000 V | HV-IGBTs |
| INGECON WIND MV100 | NPC BTB | 5–15 MW | 3.3 kV | HV-IGBTs |
| Siemesns HVDC plus | M$^2$C | – | 13.2–13.8 kV | HV-IGBTs |

*5.4. Trends in Power Converters for Multi-MW WECSs*

Most existing WECSs are equipped with low-voltage power converters. Therefore, multi-MW WTs imply high current magnitudes that decrease efficiency and increase cost and cable sizing. Consequently, novel medium-voltage power converters have been proposed in addition to the topologies above-mentioned. Modular Multilevel Cascade Converters (MMCC) are connected in parallel to form multiple cells, eliminating lower voltage harmonics and electromagnetic inference. In addition, a fault ride-through capacity is enabled due to its modular structure. Some of the MMCCs in the literature can achieve direct AC/AC conversion, which is desirable in modern WECS [109,110]. The conversion of AC/AC use MMCCs has two methods. First, the three-phase generator voltage is converted to grid voltage with a frequency of 50 Hz. Second, three single-phase voltages to three-phase grid voltage with a line frequency of 50 Hz.

### 5.4.1. Modular Multilevel Back-to-Back Converters

In this configuration, two Modular Multilevel Converter ($M^2C$) is connected by their DC ports to enable AC / AC conversion. Each converter is made of clusters and is cascaded with one inductor. Bidirectional choppers and a flying capacitor are utilized to form power cells. Of all the MMCC topologies, the $M^2C$ BTB converter is currently available on the market [111]. The $M^2C$ has been proposed for multi-MW WECS [112,113]. Figure 7a shows $M^2C$. The $M^2C$ has some difficulties when applying low speed and high torque. In addition, $M^2C$ is not a direct conversion from AC/AC and requires a BTB converter, increasing the size of the converter.

### 5.4.2. Modular Multilevel Matrix Converter

The Modular Multilevel Matrix Converter ($M^3C$) is an AC/AC converter capable of achieving high voltage levels by utilizing the series connection of full-bridge modules. The $M^3C$ has been proposed for multi-MW WECS [114,115]. It has several advantages, including modularity, flexible control, and high-voltage operation. Moreover, especially in offshore applications, reliability is a crucial parameter, and it has been addressed by $M^3C$ topology with reduced transformer size. The $M^3C$ has been proposed for the variable operation of WTs [116,117]. However, the converter branches did not have inductors, and each branch was controlled as a voltage source instead of the modern control system where each branch is controlled as a current source.

The $M^3C$ is an excellent alternative for high-power applications. The main advantages are low harmonic distortion in the grid and machine side currents due to high effective switching frequency. In addition, the $M^3C$ has several technical benefits such as better voltage range and adequate fault ride-through performance [118]. Also, the fault redundancy is improved even though the machine is under low rotational speed [119]. In the latter case, the machine currents are relatively large, but the machine back-emf is small. From this, it is found that the power oscillations (at twice the machine electrical frequency) are reduced, and the simple control system is enough to control voltage fluctuations in capacitors [118].

The operation of the converter at high power is quite difficult, as is overheating. However, the $M^3C$ could perform a stable operation at this operating point. Therefore, an $M^3C$-based WECS is designed to achieve rated output power even though the machine operates at least 5–10% below the grid frequency. In addition, the $M^3C$ is utilized for controlling multi-channel generators [120]. In this topology, it is possible to improve the fault-redundancy capability of the entire system. An $M^3C$ has been proposed for DFIG-based WECS and depicted in Figure 7b. Further, the $M^3C$ has been proposed to control DFIG-based WECSs, with appropriate low voltage ride-through performance [121]. The $M^3C$ can introduce a large voltage into the DFIG rotor and maintain the current controller; at the same time, the demagnetization of the machine is achieved by ignoring the utilization of crowbars.

### 5.4.3. Hexverter

The Hexverter performs AC/AC conversion, composed of six clusters equipped with full-bridge power cells, and it can be analyzed as a six-clusters $M^3C$. Figure 7c illustrates the hexverter. The Hexverter has been proposed as an alternative for high-power WECSs [122]. Compared to the $M^3C$, this converter has a 33% lower power cell requirement. However, to ensure proper steady-state operation, the power transfer among adjacent clusters has to be compensated by injecting an adjacent-power component that leads to an oversized design, underrated efficiency, and undesired effects on the generator.

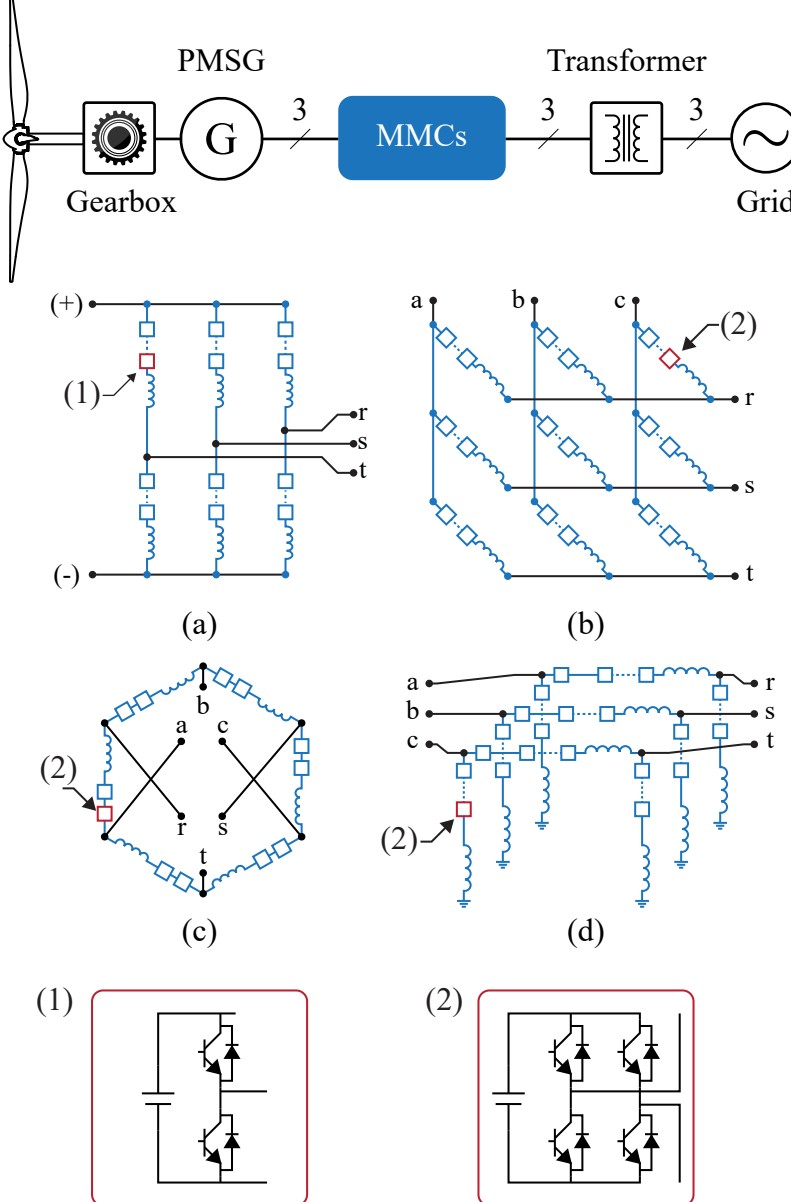

**Figure 7.** (**a**) M$^2$C. (**b**) M$^3$C. (**c**) Hexverter. (**d**) Shunt-series.

### 5.4.4. Shunt Series Modular Multilevel Converter

The WECS is equipped with shunt series modular multi-level converter for medium voltage AC grid [123]. The circulating current of the converter is identified by the control system that is used to control the average value of the floating capacitor without affecting the input and output. This topology could offer a proper solution subject to several cell and semiconductor requirements. Figure 7d shows the shunt series modular multilevel converter.

### 5.4.5. Comparison of Different Converter Topology

This section compares the most popular converter topologies for multi-MW WECS in the power range of 3 to 10 MW and above. The following power converters are selected for this comparison: parallel two-level BTB converter, current source BTB, NPC BTB converter, Hexverter, M$^2$C, and M$^3$C. Table 6 shows the performance evaluation of converters based on power rating, typical voltage, technological status, system reliability, grid code compliances, circulating current, and commercial examples. The current source BTB, M$^2$C, Hexverter and

M$^3$C are utilized for 10 MW WECSs. The semiconductor demand for the M$^2$C, Hexverter, and M$^3$C are the same. However, the DC capacitor requirements for M$^2$C are twice that of the Hexverter [124]. Furthermore, the operation of the M$^2$C at low frequencies demands the injection of common-mode voltages and higher circulating currents, which increases the size of the converter. Wind turbine downtime highly relies on generator and power converter failure [125]. Therefore, power converters with modular structures are preferred [126]. Reliability and grid code compliance performance are best in modular multilevel converters. In the future, the Hexverter and M$^3$C will be the predominant solution for multi-MW WECSs with a power rating of more than 10 MW.

**Table 6.** Comparison of different power converter topologies for WTs.

| Parameters | Parallel 2L BTB Converter | Neutral-Point Clamped BTB Converter | Current Source BTB Converter | M$^2$C | Hexverter | M$^3$C |
|---|---|---|---|---|---|---|
| Power rating | 0.75–6 MW | 3–8 MW | 3–10 MW | 10 MW and above | 10 MW and above | 10 MW and above |
| Typical Voltage | LV | LV | LV | LV and MV | LV and MV | LV and MV |
| Technology Status | Well Established | Well Established | Research Only | Research Only | Research Only | Research Only |
| Reliability of system | High | Medium | High | High | High | High |
| Grid Code Compliance | Medium | Good | Good | Excellent | Low | Excellent |
| Circulating Currents | Medium | – | – | High | High | Low |
| Commercial Example | Ingeteam FC LV | Ingeteam FC MV, Converteam 7000 | Rockwell PL 7000 | Siemens HVDC plus | – | – |

## 6. Conclusions

This article presents a comprehensive study of multi-MW WT generators and power converters. Currently, WT can easily surpass the 10 MW barrier with nominal power up to 16 MW and a rotor diameter of 250 m. The NPC BTB converter, equipped with medium voltage semiconductors, is the most commonly used power converter in multi-MW WT, but new topologies based on modular structures have been indicated as potential solutions for the next generation of large WTs. This review has drawn the following conclusions.

- The DD PMSGs are highly preferable generators for high-power WECSs, whereas these generators are associated with REM, which could increase the cost, size and mass of the generators.
- The HTS generators can lead to the most significant weight and size reductions. However, the initial cost of this technology is still an issue to solve before reaching a higher technology readiness level.
- The LTS and MgB$_2$ superconducting generators are under conceptual level. Therefore, there is an opportunity to explore these generators for the high-power wind industry.
- Currently, low-voltage power converters are highly dominating the wind industry. However, the reliability of those converters is a critical issue, and it needs to be addressed in future.
- This study suggests that MMC converters, such as Hexverter and M$^3$C, could be an appropriate future solution for WTs above 10 MW operating at the MV level as these converters have high power density, fault tolerance, modularity and high power quality.

In the future, massive enlargement of high-power WECSs will be available worldwide. However, the technological development of WECS will play a significant role in wind power systems. Therefore, the following future studies could be beneficial for further development in WECSs.

- The HTS generator is the alternative solution to replace DFIG and PMSG [16]. However, the superconducting generators are still in the process of concept level. Currently, the AMSC manufactures the HTS generator, and replacing HTS with $MgB_2$ could reduce the total cost [70]. Therefore, further studies are needed in this area to expand superconducting generators.
- WECS downtime is strongly dependent on power converter failures. Therefore, the reliability of power converters is a challenging area for future research, and medium-voltage power converters could improve the reliability issues.

**Author Contributions:** Conceptualization, S.R., M.D. and R.C.; methodology, R.C. and J.R.; writing: S.R., M.D. and E.E.; supervision, R.C. and J.R., visualization, E.C.; review and editing: S.R., M.D. and E.E. All authors have read and agreed to the published version of the manuscript.

**Funding:** This work was funded by the Agencia Nacional de Investigación y Desarrollo (ANID) of Chile, under projects FONDECYT Post Doctoral Project N° 3200934, FONDECYT N° 11191163 and FONDEQUIP EQM-200234.

**Institutional Review Board Statement:** Not applicable

**Informed Consent Statement:** Not applicable

**Data Availability Statement:** Not applicable.

**Conflicts of Interest:** The authors declare no conflict of interest.

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
