# Peer review of "A Review of Generators and Power Converters for Multi-MW Wind Energy Conversion Systems"

_processes, doi:10.3390/pr10112302_

Round 1
Reviewer 1 Report
This study focuses on the current trends in generators and power converters solely used for multi–MW WTs (above 6 MW).
It presents a detailed study on multi-megawatt wind energy conversion systems focusing on a review of different generators and power converters commercially available in the market and new technologies that have been proposed in the recent literature for high-power ratios.
Overall, the introduction offers an accurate, general description of the subject for readers to understand the overall scope of the study while giving them comprehensive information on the problem and its rationale.
It seems to me that the author put his main efforts into his main point of emphasis on the multi-megawatt wind energy conversion systems, and hence it is more applicable to the subject of his findings deal with.
The authors present an analysis of the technical and scientific literature on wind energy conversion systems, providing a comparison with traditional systems and providing pros and cons from all points of view, not only technical but also economic.
The literature cited is relevant to the study. The conclusion is well structured since it includes a discussion of the findings obtained and uses straightforward and concise language to summarize the research subject for the reader.
To improve the strength of the research, I suggest some comments that should be taken into consideration:
- Since the study is a review study, I, therefore, suggest adding a word suffix to the title "- A review".
- The aim of the work is not specified in the abstract. Results are also unclear in the abstract section.
- In line 134, you mentioned doubly fed induction generators as DFIMs, while you refer to them as DFIG in the same line, please address this issue.
- The use of lumped references as in [11-15] should be avoided to provide a sufficient representation of the main contribution of each referenced paper.
- Do not start the phrase with the reference number of the cited paper (e.g., „In [11,13,19] …"
- Abbreviations that appear for the first time must carry the full name.
- The manuscript needs Future outlook and recommendations
Author Response
Comment 1: Since the study is a review study, I, therefore, suggest adding a word suffix to the title "- A review".
Reply: The title of the article has been modified in the revised manuscript. Thank you for the suggestion.
Comment 2: The aim of the work is not specified in the abstract. Results are also unclear in the abstract section.
Reply: The abstract has been revised according to the suggestions.
Comment 3: In line 134, you mentioned doubly fed induction generators as DFIMs, while you refer to them as DFIG in the same line, please address this issue.
Reply: The doubly-fed induction generator is referred to as DFIG in the revised manuscript.
Comment 4: The use of lumped references as in [11-15] should be avoided to provide a sufficient representation of the main contribution of each referenced paper.
Reply: The main contribution of each reference from [11] to [15] has been mentioned in the revised manuscript.
Comment 5:Do not start the phrase with the reference number of the cited paper (e.g., „In [11,13,19] …"
Reply: According to the suggestion, the phrases starting with the reference number have been revised.
Comment 6:-Abbreviations that appear for the first time must carry the full name.
Reply: This has been taken care in the revised manuscript.
Comment 7: The manuscript needs Future outlook and recommendation
Reply: In this revised version of the paper, future studies related to generators and power converters have been added at the end of the conclusion section.
Reviewer 2 Report
The language of the article needs revision
The abstract is very brief considering that the study is in the form of review articles
It is necessary to add the future studies section at the end of the conclusion
Considering that the study is in the form of review articles, it is necessary to explain about the control methods according to the Fig. 2.
Regarding the types of power converters, according to the topic of the article, it is necessary to present and explain.
Many studies are reviewed in this article, but their advantages and disadvantages are not presented. Studies need to be categorized in a table based on advantages and disadvantages.
In this study, there is no explanation about power system indicators considering wind units in the power system network.
Author Response
Comment 1: The language of the article needs revision.
Reply: This has been taken care in the revised manuscript.
Comment 2: The abstract is very brief considering that the study is in the form of review articles
Reply: According to the suggestion, the abstract has been modified in the revised manuscript.
Comment 3: It is necessary to add the future studies section at the end of the conclusion
Reply: Future studies related to generators and power converters have been added at the end of the conclusion section.
Comment 4: Considering that the study is in the form of review articles, it is necessary to explain about the control methods according to the Fig. 2.
Reply: We thank the reviewer for this comment. Accordingly, the beginning of Section 2 was modified to explain the control method for converter, WECS and grid control. Please check pages 3 and 4 for more details.
Comment 5: Regarding the types of power converters, according to the topic of the article, it is necessary to present and explain.
Reply: We thank this comment, and we have improved Section 5. In this revised version of the paper, power converters for multi-MW WECS have been categorized as available in the industry and as technological trends proposed in recent literature. Please check pages 13 to 17.
Comment 6: Many studies are reviewed in this article, but their advantages and disadvantages are not presented. Studies need to be categorized in a table based on advantages and disadvantages.
Reply: In this revised version of the paper, we have included the pros and cons of generators and power converters for WECS. On the one hand, the generator part has been compared in Table 1, page 9. On the other hand, power converters are compared in Table 6.
Comment 7: In this study, there is no explanation about power system indicators considering wind units in the power system network.
Reply: We agree about the importance of power system indicators for WECS, and we have included a brief comment in the introduction about this line of thought (see references [19] to [22]). However, we’d like to state that our main scope is the analysis of generators and power converters used in WECS, from a power electronics point of view, excluding the power system perspective.
Reviewer 3 Report
The present paper aims to present a comprehensive survey of multi-MW WT generators and power converters. Their generator-converter configurations have evolved to provide reliability, power density, maintainability, efficiency, and robustness. Thus, the paper makes a detailed study of multi-megawatt wind power conversion systems with a focus on a review of different commercially available commercially available generators and power converters and new technologies that have been proposed in recent literature for high power ratios. Currently, TWs can easily surpass the 10 MW barrier, reaching a power rating of 16 MW and a rotor diameter of 250 m. In this power picture, the PMSG seems to be the preferred generator, usually connected directly to the turbine to avoid the gearbox in order to increase power density. The BTB NPC converter, equipped with medium voltage semiconductors, is the most widely used power converter in multi-MW WT. NPC converters used in WTs are equipped with LV and MV IGBTs, depending on the power and voltage level of the applications. DD PMSGs are highly preferable generators for high power WECSs, while these generators are associated with REM, which can increase the cost, size, and mass of the generators. There are still research challenges related to the size and mass
deduction in DD PMSGs. HTS generators may lead to the most significant weight, and size reductions. However, the initial cost of this technology is still an issue to be solved before reaching a higher level of technology readiness. The LTS and MgB2 superconducting generators are at a conceptual level. Therefore, there is an opportunity to exploit these generators for the high-power wind industry. The paper is well written and well formulated with new contributions. Therefore, the article is accepted for publication.
Author Response
General comment: The present paper aims to present a comprehensive survey of multi-MW WT generators and power converters. Their generator-converter configurations have evolved to provide reliability, power density, maintainability, efficiency, and robustness. Thus, the paper makes a detailed study of multi-megawatt wind power conversion systems with a focus on a review of different commercially available commercially available generators and power converters and new technologies that have been proposed in recent literature for high power ratios. Currently, TWs can easily surpass the 10 MW barrier, reaching a power rating of 16 MW and a rotor diameter of 250 m. In this power picture, the PMSG seems to be the preferred generator, usually connected directly to the turbine to avoid the gearbox in order to increase power density. The BTB NPC converter, equipped with medium voltage semiconductors, is the most widely used power converter in multi-MW WT. NPC converters used in WTs are equipped with LV and MV IGBTs, depending on the power and voltage level of the applications. DD PMSGs are highly preferable generators for high power WECSs, while these generators are associated with REM, which can increase the cost, size, and mass of the generators. There are still research challenges related to the size and mass deduction in DD PMSGs. HTS generators may lead to the most significant weight, and size reductions. However, the initial cost of this technology is still an issue to be solved before reaching a higher level of technology readiness. The LTS and MgB2 superconducting generators are at a conceptual level. Therefore, there is an opportunity to exploit these generators for the high-power wind industry. The paper is well written and well formulated with new contributions. Therefore, the article is accepted for publication.
Reply: We thank the reviewer for his/her positive comments.
Round 2
Reviewer 2 Report
Requested corrections have been made and the quality level of the article has improved.